# Adequacy of endoscopic recognition and surveillance of gastric intestinal metaplasia and atrophic gastritis: A multicentre retrospective study in low incidence countries

Judith Honing[1], W Keith Tan[1], Egle Dieninyte[2], Maria O'Donovan[3], Lodewijk Brosens[4], Bas Weusten[5,6], Massimiliano di Pietro[1] *

1 Early Cancer Institute, University of Cambridge, Cambridge, United Kingdom, 2 Department of Gastroenterology, Vilnius, Lithuania, 3 Department of Pathology, University of Cambridge, Addenbrooke's University Hospital, Cambridge, United Kingdom, 4 Department of Pathology, University Medical Center Utrecht, Utrecht University, Utrecht, The Netherlands, 5 Department of Gastroenterology and Hepatology, University Medical Center Utrecht, Utrecht University, Utrecht, The Netherlands, 6 Department of Gastroenterology and Hepatology, Sint Antonius Hospital, Nieuwegein, The Netherlands

* md460@cam.ac.uk

## Abstract

### Background

Gastric atrophy (GA) and gastric intestinal metaplasia (GIM) are precursor conditions to gastric adenocarcinoma (GAC) and should be monitored endoscopically in selected individuals. However, little is known about adherence to recommendations in clinical practice in low-risk countries.

### Objective

The aim of this study was to evaluate endoscopic recognition and adequacy of surveillance for GA and GIM in countries with low GAC prevalence.

### Methods

We retrospectively analysed patients diagnosed with GIM or GA in three centers in The Netherlands and UK between 2012 and 2019. Cases with GIM and/or GA diagnosis at index endoscopy were retrieved through systematic search of pathology databases using 'gastric' and 'intestinal metaplasia' or 'atrophy' keywords. Endoscopy reports were analysed to ascertain accuracy of endoscopic diagnoses. Adequacy of surveillance was assessed following histological diagnosis at the index endoscopy based on ESGE guidelines published in 2012.

### Results

We included 396 patients with a median follow-up of 57.2 months. Mean age was 66 years and the rates of antrum-predominant versus extensive GIM were comparable (37% vs

**Data Availability Statement:** All relevant data are within the paper and its Supporting information files.

**Funding:** The author(s) received no specific funding for this work.

**Competing interests:** The authors have declared that no competing interests exist.

38%). Endoscopic recognition rates were 48.5% for GA and 16.3% for GIM. Surveillance was adequately carried out in 215 of 396 patients (54.3%).

## Conclusion

In countries with a low incidence of GAC, the rate of endoscopic recognition of gastric pre-cancerous lesions and adherence to surveillance recommendation are low. Substantial improvement is required in endoscopic training and awareness of guidelines recommendation in order to optimise detection and management of pre-malignant gastric conditions.

## Introduction

Gastric adenocarcinoma (GAC) has a poor prognosis with an overall 5-year survival lower than 20% [1]. While the worldwide incidence of gastric cancer is declining [1,2], alarmingly there seems to be a trend showing an increased incidence in younger age groups (under 50) [3,4]. It is therefore essential to remain vigilant and focus on ways to prevent this lethal disease.

The main reason for its dismal prognosis is that patients typically present at advanced stages of disease. Gastric atrophy (GA) and gastric intestinal metaplasia (GIM) are histopathological stages of the chronic atrophic gastritis and precursor conditions for GAC, the most common pathological variant of gastric cancer, which arises from a background of either H.pylori infection or auto-immune gastritis [5]. Recently, studies have shown that 6–8% of all upper gastrointestinal (GI) cancers occur within three years of a negative esophago-gastro-duodenoscopy (EGD) [6,7]. This type of cancer, also known as post-endoscopy upper GI cancer (PEUGIC), tends to be smaller in size at diagnosis [8]. Moreover, there is evidence that GA and GIM are also often overlooked at endoscopy [9].

The 2012 European Society of Gastrointestinal Endoscopy (ESGE)MAPS guidelines on the management of precancerous lesions in the stomach recommend surveillance for patients with extensive GA and/or GIM, and for patients with GA/GIM of the antrum-predominant stomach (distal) in the context of persistent H.pylori infection or a positive family history of gastric cancer [10]. Distal GA or GIM, in the absence of additional risk factors, is considered a low-risk condition and therefore not an indication for surveillance. The introduction of this guideline seems to have had a positive effect on detection of early stage T1 cancers and survival outcome in some European countries [2], highlighting the importance of early recognition and surveillance.

The current gold standard for detection of GA and GIM is the histopathological diagnosis on biopsies taken during diagnostic EGD, generally performed to investigate abdominal complaints or anaemia. This diagnosis relies on the ability of the endoscopist to identify pre-malignant stomach lesions, since gastric biopsies are not taken routinely. However, recognition of GA and GIM is challenging, therefore impacting on the diagnostic accuracy of the gold standard [9]. This could be particularly the case in Western European countries, where the prevalence of GA and GIM is much lower than Eastern European and Asian countries [11,12]. Furthermore there is limited evidence about the rate of endoscopic recognition rates in Western countries. Therefore, the aims of this study were *i.* to quantify the rate of endoscopic recognition of GA and GIM in a real-life clinical practice using histopathologic diagnosis as gold standard and *ii.* to investigate the adherence to surveillance recommendations in two countries with a low incidence of GAC.

## Material and methods

### Patient selection

This was a retrospective study conducted in one academic hospital in the UK (center 1), one academic (center 2) and one non-academic teaching hospital in the Netherlands (center 3). Records were searched for all hospital between 2012–2019 after implementation of the revised MAPS guidelines in 2012 [10]. Patients' records were retrieved using a search of the pathology records. In the UK center, a search was performed of the pathology records using the search terms 'gastric' and 'intestinal metaplasia' or 'atrophy'. In the Netherlands, data were retrieved using PALGA: the nationwide network and registry of histo- and cytopathology in the Netherlands [13]. The following PALGA search criteria were used: gastric (T63), intestinal metaplasia (M73320), or metaplasia (M73000), or atrophic gastritis (M58010) or atrophy (M58000) (https://www.palga.nl/palga-on-line-thesaurus.html). Ethics committee approval was sought, but since the retrospective nature of the study it was not deemed to require authorisation by the Ethics committee and no informed consent needed to be obtained. However, all patient records were verified for any objections against the use of their data for retrospective analysis. The study adhered to the national data protection regulations. The anonymized dataset is included as (S1 Dataset).

### Definitions

Distal GA and/or GIM was defined as sparse foci of atrophy and/or intestinal metaplasia, respectively, involving mucous secreting glands of the antrum and/or incisura only. Proximal GA and/or GIM was defined foci of atrophy and/or intestinal metaplasia involving the fundus and/or body. Extensive (pangastric) GA and/or GIM was defined by extensive atrophic or intestinal metaplastic lesions involving both proximal and distal stomach. In a small proportion of cases where GA and/or GIM was suspected on histopathology, but the diagnosis remained indeterminate, these patients were designated a diagnosis of GA and/or GIM unless this was excluded at follow-up EGD. H.pylori infection was determined on the pathology report of the index endoscopy; further search on historical data regarding a previous H.pylori infection was not performed. Endoscopic recognition of GA was defined as endoscopic diagnosis of gastric atrophy in cases with GA with or without GIM. Endoscopic diagnosis of GIM was defined as endoscopic diagnosis of GIM in cases with histopathologic evidence of GIM. Data regarding family history was retrieved from the medical records; family history was regarded as positive or negative if the physician specifically mentioned this item in the consultation record. Data was regarded missing if no data could be retrieved from the medical records and it was unclear if this item was discussed at the time of consultation.

### Inclusion and exclusion criteria

Patients were included if GA or GIM was found on an index endoscopy between 2012–2019. Exclusion criteria were MALT lymphoma or gastric cancer at index endoscopy, previous gastric cancer or gastric dysplasia requiring surveillance, referral for endoscopic treatment of dysplasia. Only cardia biopsies without any fundus or body biopsies were not considered as adequate sampling and therefore cases with isolated cardia IM in the absence of non-cardia gastric biopsies were excluded.

### Endoscopy

All endoscopy reports were manually reviewed. If the endoscopist made note of the suspicion of GA or GIM this was regarded as adequate recognition. Non-specific terminology such as presence of gastritis, nodular patches or erythema were not regarded as adequate. Biopsy

location was classified according to the following criteria: i. proximal biopsies if taken at the fundus and/or body, ii. distal biopsies if taken in the antrum or at the incisura, iii. complete sampling if biopsies at both locations, iv. unknown if the location was not specified in the endoscopy report. The biopsy protocol was regarded adequate if at least one proximal and one distal biopsy were taken to determine the extent of GA and/or GIM. We included endoscopies performed by all grades of endoscopists accredited to operate independently, including consultants, trainees and nurse endoscopists.

## Surveillance

To determine adequacy of surveillance we used the revised 2012 ESGE MAPS guideline criteria. Since in 2019 both the British Society of Gastroenterologists (BSG) and the European Society of Gastroenterologists (ESGE) updated their guidelines, patients from 2019 were scored according to this revised guideline. Adequacy of surveillance was based on topographic extent of the disease, presence of risk factors and completeness of endoscopic sampling. Risk factors were persistent H. pylori infection and family history of gastric cancer. Based on the MAPS criteria, surveillance was regarded adequate if:

1. initial follow up EGD was requested in cases of extensive GA or GIM diagnosed in proximal gastric biopsies, or GA/GIM at any location in a patient with known risk factors;

2. no surveillance was indicated for cases of distal GA and/or GIM with negative proximal biopsies, and in the absence of clinical risk factors.

   Adherence to surveillance was regarded inadequate in the following cases:

1. extensive atrophy or proximal IM not followed up;

2. distal IM with known risk factors not followed up;

3. distal GA and/or GIM with only distal biopsies and no follow-up endoscopy planned to determine the extent of the disease with the full Sydney biopsy protocol;

4. GA and/or GIM in biopsies taken from unknown biopsy location and no follow-up endoscopy with full Sydney biopsy protocol.

   In cases where distal GIM only was reported, and information on family history could not be retrieved based on the patient's record, lack of follow up endoscopy was regarded as adequate.
   Although the guidelines do not recommend an age limit for surveillance of 75 years, in some countries this criterium is widely adopted. Surveillance adherence was calculated for both with and without an age limit of 75 years.

## Statistics

Normality testing for continuous variables was performed using Shapiro-Wilk test. Differences in proportions between groups were compared using the Chi-square and Fisher exact test, where appropriate. The Kruskal Wallis test was used to compare the median age difference between groups. For all outcomes, a 2-sided p-value of $<0.05$ was denoted as statistical significance. All analyses were performed in IBM SPSS package 28 (Armonk, New York, IBM Corp).

## Results

### Patient cohort

We included 396 patients with a histopathological diagnosis of GA and/or GIM, of which 313 were from the two academic centers and 83 were from a non-academic teaching hospital

**Table 1. Patient demographics.**

| Variable | Centers | | | | p-value |
|---|---|---|---|---|---|
| | **All** | **1** | **2** | **3** | |
| | **(n = 396)** | **(n = 100)** | **(n = 213)** | **(n = 83)** | |
| Age, median (IQR) | 68 (19) | 70 (17) | 67 (19) | 65 (18) | 0.004[$]I |
| Sex (% female) | 220 (56) | 58 (58) | 105 (49) | 57 (69) | 0.009[%] |
| Intestinal metaplasia, n (%) | | | | | 0.003[&] |
| • None | 59 (15) | 21 (21) | 36 (17) | 2 (2) | |
| • Proximal | 73 (18) | 27 (27) | 28 (13) | 18 (22) | |
| • Distal | 145 (37) | 11 (11) | 90 (42) | 44 (53) | |
| • Both locations | 80 (20) | 19 (19) | 46 (22) | 15 (18) | |
| • Present, location unknown | 39 (10) | 22 (22) | 13 (6) | 4 (5) | |
| Helicobacter Pylori, n (%) | | | | | |
| • active infection | 59 (15) | 13 (13) | 35 (16) | 11 (13) | 0.617[%] |
| • history of eradication | 50 (13) | 11 (11) | 32 (15) | 7 (8) | 0.263[%] |
| Family history of gastric cancer, n (%) | | | | | <0.001[&] |
| • Yes | 17 (4) | 8 (8) | 6 (2) | 3 (3) | |
| • No | 187 (47) | 34 (34) | 120 (56) | 33 (40) | |
| • Unknown | 192 (49) | 58 (58) | 87 (41) | 47 (57) | |

[$]Kruskal-Wallis test;

[%]Chi-squared test;

[&]Fisher-exact test.

IQR, interquartile range, Center 1 and 2 academic centers, center 3 non-academic hospital.

(Table 1). The median age was 68 years (IQR 19), indicating most patients would qualify for surveillance based on their age. In 15% of patients, only GA without GIM was found. The rates of localized versus extensive GIM were comparable, with 37% patients having distal GIM only, and 38% having extensive GIM (20% pangastric and 18% proximal), and in 10% the location was unknown. In a small proportion (4.3%) of patients, there was a positive family history of gastric cancer, although 49% of the patients did not have this information recorded. The mean follow-up was 57.2 months (standard deviation, SD 33.3).

## Endoscopic recognition

The rate of endoscopic recognition of GIM was low, with a sensitivity of only 16.3% (Fig 1, S1 Table). As expected, antral IM was more easily recognized compared to proximal IM. In cases with either antral IM or with pangastric IM, recognition rates were 22.6% (50/225) whereas in patients with proximal IM only (5/73) this rate was only 6.8%. One academic centre performed better compared to the other academic centre and regional hospital. However, these differences were less apparent for GA, which was generally better recognized in 48.5% of the cases (61.1% and 40.7% for the academic centers and 44% for the district general hospital, Fig 1). Overall endoscopic recognition of any premalignant condition by endoscopists in the presence of GA or GIM was 42.2% (S1 Table).

## Adequacy of surveillance

In total 14.4% (57/396) received surveillance and 22.4% (89/396) no surveillance according to the guidelines, whereas another 17.4% (69/396) were discharged because of age (Table 2).

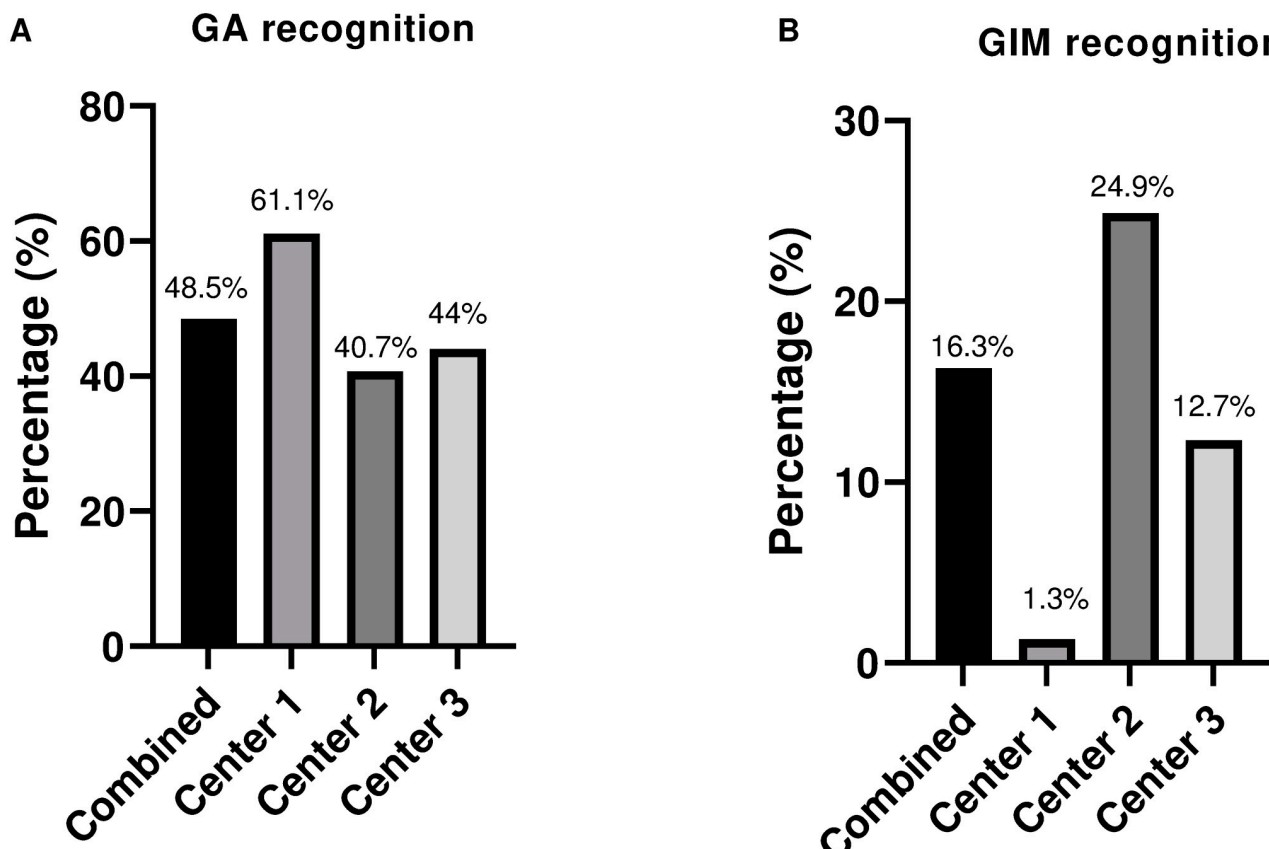

**Fig 1. Endoscopic recognition of GA and GIM.** (A) Rate of endoscopic recognition of GA with or without GIM by the endoscopist. (B) Rate of endoscopic recognition of GIM. Center 1 and 2 are academic, center 3 is a non-academic teaching hospital.

Adherence to surveillance guidelines was adequate in 54.3% (215/396) when including an age cut-off of 75, and 36.9% without age limit (Fig 2) and was similar across centers (S2 Table).

Adequacy of surveillance was partly determined by the acquisition of both distal and proximal biopsies. In the majority of patients (68%, 268/396) biopsies from both regions were taken, but in 15% only distal biopsies were taken and in 9% only proximal (S3 Table). From another 8% the biopsy location could not be established.

**Table 2. Adequacy of Surveillance of GIM per gastric region.**

| Variables | No GIM (n = 59) | Proximal GIM (n = 73) | Distal GIM (n = 145) | Pangastric GIM (n = 80) | GIM unknown location (n = 39) | Total (N = 396) |
|---|---|---|---|---|---|---|
| Adequate surveillance, n (%) | 8 (13.6) | 14 (19.2) | 15 (10.3) | 16 (20) | 4 (10.2) | 57 (14.4) |
| Inadequate, n(%) | 36 (61.0)$ | 46 (63.0) | 34 (23.5) | 43 (53.8) | 22 (56.4) | 181 (45.7) |
| No surveillance per guidelines, n (%) | 1 (1.7) | 3 (4.1) | 81 (55.9) | 3 (3.8) | 1 (2.6) | 89 (22.5) |
| No surveillance due to age*, n (%) | 14 (23.7) | 10 (13.7) | 15 (10.3) | 18 (22.5) | 12 (30.8) | 69 (17.4) |

*>75 years.

$patients with gastric atrophy but no surveillance.

GIM: Gastric intestinal metaplasia.

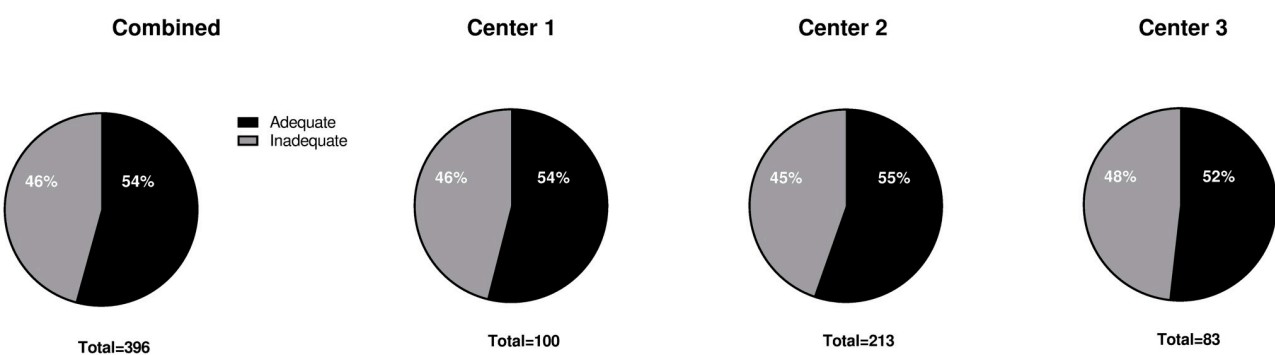

**Fig 2. Adherence to surveillance.** Percentages of adherence to surveillance guidelines. Adequate surveillance was regarded as either surveillance initiated or surveillance withheld based on the ESGE/MAPS guideline criteria and including an age limit of 75 years. Center 1 and 2 are academic centers, center 3 is a non-academic teaching hospital.

## Progression to dysplasia/cancer and missed dysplasia/cancer at index endoscopy

Overall one patient progressed towards cancer in our cohort. This case was a 72-year-old male initially diagnosed during an endoscopic ultrasound requested for side branch intraductal papillary mucinous neoplasm (IPMN) surveillance, where patches of suspected body IM were biopsied. Two years later the patient underwent another EGD where atrophic signs and erosions in the antrum were reported but not biopsied. After both endoscopies no surveillance was initiated. Three years later (age 77 years), the patient was admitted to the hospital with melena and EGD revealed a large antral ulcer with intestinal-type GAC. CT showed widespread liver and peritoneal metastasis. Due to poor performance status the patient received one cycle of palliative radiotherapy but died one month later.

In another 3 cases, a dysplastic lesion or an early cancer was missed at index endoscopy. There were 2 cases where initial random biopsies on index endoscopy showed LGD, which was not identified as a lesion on EGD. In one patient the follow-up treatment with EMR revealed a pT1a tumor. The other case with LGD was lost to follow-up, but repeat endoscopy 5 years later showed a persisting lesion. This lesion was removed en-bloc with an endoscopic mucosal dissection and the histopathology showed LGD with clear margins. Due to his age (83 years) and comorbidities the patient was discharged afterwards but remains well one year later.

The third case is a patient with a missed cancer revealed at early repeat endoscopy within 12 months from baseline procedure. On the first endoscopy the biopsies showed GA with GIM but no follow-up was initiated. Due to persistent symptoms, a second EGD within one year was requested which showed GAC and the patient was treated curatively with a gastrectomy (pT1bN0).

## Discussion

This study demonstrates that endoscopic recognition of the pre-malignant stomach even in tertiary centres is suboptimal and that adherence to guidelines for surveillance of GA and GIM is low with little variation among centres.

Endoscopic recognition of gastric premalignant disease is essential to initiate a correct surveillance strategy. We showed that the rate of endoscopic recognition of GA and GIM is lower than 50% with sensitivity of 48.5% for GA and 16.3% for GIM. When looking at variability among centres, one of our institutions had a significantly higher rate of endoscopic

recognition of GIM (24.9% vs 12.3% and 1.3%), but this is likely related, at least in part to the different rates antral GIM, which is more easily recognized by the endoscopist. Overall, our results are in keeping with previous data on endoscopic recognition from moderate-high prevalence countries where rate of endoscopic recognition was similarly low [9,14]. In a single-center study in Korea, the sensitivity and specificity of endoscopic GA regardless of IM in the antrum was 61.5% and 57.7% respectively, and of the body 46.8% and 76.4%. However, in this cohort all procedures were performed by one endoscopist and a large proportion had previous GAC, which gives a selection bias for detection of GA and/or GIM and is not representative of the standard referral population [14]. Another recent retrospective cohort study from Poland, which has high prevalence of H.pylori infection, including 3000 patients referred with clinically suspected gastritis reported an endoscopic recognition rate of IM of 19.4% (128/660) [9]. Sensitivity and specificity for endoscopic recognition of GA were 69.5% and 69.5% and of IM 19.4% and 97.9% respectively. However, in only 31.1% of the patients, biopsies from both antrum and corpus were taken, whereas in our study this was 68% [9]. In a study from Greece, similar sensitivity and specificity for GIM were reported (74.6% and 95%), but in this study a limited number of biopsies was allowed with a minimum of 3 biopsies form corpus and antrum which could have led to an under representation of the actual GIM [15].

While it seems surprising that the level of GIM recognition with our current imaging-enhancing techniques is lower than GA recognition, this finding is in keeping with a previous report [9]. In our cohort in most cases the endoscopist did not recognize IM, but only suspected an inflammatory process, which triggered biopsies revealing IM. We suspect that, due to the limited knowledge of the disease spectrum of GA progressing towards GIM and eventually dysplasia, endoscopists may be unaware of the recommendations to make use of NBI to thoroughly inspect the gastric mucosa to identify GIM and possible dysplastic lesions [10,16]. The lack of awareness of guideline recommendations in clinical practice is also reflected by adherence to surveillance protocols under 50% and inadequate biopsy sampling in 32%.

Poor recognition of a pre-malignant stomach disease can lead to reduced level of alert for subtle malignant lesions already present at index endoscopy and subsequently missed cancer diagnosis. Recent evidence in European cohorts shows that 6–7% of upper gastrointestinal cancers are missed at a previous endoscopy [6,7]. In a small retrospective cohort study from US which analysed 91 patients with gastric cancer, the majority of the cases with a high-risk condition at previous endoscopy did not receive follow-up, indicating a failure to initiate surveillance strategy, which could lead to GAC [17]. Our data support this growing evidence of missed upper GI diagnosis on EGD and calls for diagnostic quality indicators to improve endoscopists performance.

To increase GA,GIM and subsequent neoplastic lesion detection, several steps in education and monitoring will be required. The first priority should be training of endoscopists, which starts with awareness of our knowledge gap. To acquire better recognition skills, training modules could be implemented in either specialty training or using specialist educational programs. Educational programs to increase adenoma detection rate (ADR) in colonoscopy were shown to be effective with sustained effect over a longer follow-up period [18]. A recent interventional study on upper GI quality indicators showed a training session on the current quality indicators for upper GI endoscopy led to increased adherence to guidelines recommendations [19]. Although GIM detection rates did not increase after the intervention, GIM detection rates did increase with prolonged inspection time during EGD suggesting longer time slots are required [19]. This is comparable to a study in the field of Barrett's esophagus where adherence to Seattle protocol increased after an educational program intervention [20].

Alternatively, detection rates could improve with the use of image-enhanced endoscopy. Narrow band imaging (NBI), which is a more routinely available modality can increase the detection of GIM in patients under surveillance and increased dysplasia detection [21,22]. The OLGIM classification is a commonly used pathological grading system to classify the extent of GIM, and higher extent is associated with a higher risk of neoplastic progression [23]. In a recent multicentre study, the use of an endoscopic NBI-based grading system to estimate the extent of GIM and targeted the biopsies led to high diagnostic accuracy for OLGIM III/IV with an area under the ROC curve of 0.96 (95% confidence interval [CI] 0.93–0.98) [24]. Although the OLGIM classification might be an interesting prognostic factor, the classification is currently not widely used in centres were the study was performed and therefore we could not include this data in the study. Novel tools such as artificial intelligence could in the future provide both endoscopists and pathologists with support in pattern recognition and aid in the diagnostic process. For example, a recently developed computer-aided detection system (CAD) showed significantly higher sensitivity for both GA and GIM (0.87 and 0.90, respectively) compared to non-expert endoscopists (0.83 and 0.74, respectively) [25]. Another recent developed digital pathology workflow for gland segmentation could aid the pathologist in analysing gastric biopsies to allow automated quantitative analysis and disease classification [26].

In the last decade there has been an increased awareness of the importance of surveillance of GA and GIM, with several updated guidelines [10,16,27] recommending surveillance in individuals with high-risk conditions. Our study shows that adherence to surveillance guidelines was suboptimal, with an inadequacy rate of 45.7%. In 31% (56/181) of the cases this was a consequence of inadequate biopsies with the lack of a repeat endoscopy for mapping, but in 49% (89 /181) of the cases with proximal or pangastric IM under the age of 75 surveillance was not initiated. A recent retrospective cohort study from US showed similar low rate of adherence to recommendation, with 55% of cases not receiving clinically indicated follow-up [28]. These low surveillance numbers likely stem from underestimation of the related risk, which can lead to preventable cancer deaths.

Our study has some limitations. Firstly, this was a retrospective study, therefore we could not establish some important clinical information, with particular reference to risk factors, and in a small proportion of patients the location of the biopsies. While it is possible that the clinician collected this information but did not report it, this information was not available during retrospective review of the cases. However, the retrospective design has the advantage of providing a snap shot of a more real-life clinical performance compared to prospective studies, eliminating the Hawthorne effect, whereby the awareness by physicians of being monitored within a trial setting alters clinical performance. Second, we did not collect data regarding the level of expertise of the endoscopist as we were unable to fully gauge the amount of supervision. Furthermore, although in some patients neoplastic progression was observed, the study was not designed to assess the neoplastic progression rate. Since many patients lacked a follow-up endoscopy our study cannot reliably address this question. Finally, in 30% of the cases biopsies were not taken according to the Sydney protocol, which leads to an incomplete histopathology diagnosis to determine the extent of the disease. This highlights the importance of taking both proximal and distal gastric biopsies, in case of GA/GIM suspicion or in case of any gastritis.

In conclusion, this study demonstrates that in three centres from countries with low incidence of GAC, rate of endoscopic recognition of GA and GIM is low and adherence to surveillance recommendation suboptimal. This should prompt future studies to investigate tools to optimize recognition and urges the development of diagnostic quality indicators to reduce post endoscopy upper GI cancers.

## Supporting information

**S1 Table. Endoscopic recognition of different histopathologic conditions.** GA: gastric atrophy GIM: gastric intestinal metaplasia.
(DOCX)

**S2 Table. Adequacy of surveillance in different centers.** Center 1: Adequate surveillance in 23 of 100 patients (23%) or 54 (54%) when including the cases without surveillance because of age. Center 2: Adequate surveillance in 82 of 212 patients (38,5%) and 118 (55,4%) when including the cases without surveillance because of age. Center 3: Adequate surveillance in 41 of 83 patients (49,4%) and 43 (51,8%) when including the cases without surveillance because of age.
(DOCX)

**S3 Table. Distribution of the biopsies based on Sydney protocol.** * refers to antrum and incisura.
(DOCX)

**S1 Dataset.**
(XLSX)

## Acknowledgments

We would like to thank the Dutch Nationwide Pathology Databank (PALGA) for their help with retrieving the pathology data.

## Author Contributions

**Conceptualization:** Judith Honing, Bas Weusten, Massimiliano di Pietro.

**Data curation:** Judith Honing, Egle Dieninyte, Maria O'Donovan, Lodewijk Brosens.

**Formal analysis:** Judith Honing, W Keith Tan, Massimiliano di Pietro.

**Methodology:** Judith Honing, W Keith Tan, Massimiliano di Pietro.

**Resources:** Maria O'Donovan, Lodewijk Brosens, Bas Weusten, Massimiliano di Pietro.

**Supervision:** Bas Weusten, Massimiliano di Pietro.

**Writing – original draft:** Judith Honing.

**Writing – review & editing:** W Keith Tan, Egle Dieninyte, Maria O'Donovan, Lodewijk Brosens, Bas Weusten, Massimiliano di Pietro.

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
