## [Decision Letter · Decision Letter 0]

9 Mar 2023

PONE-D-23-03805Adequacy of endoscopic recognition and surveillance of gastric intestinal metaplasia and atrophic gastritis: a multicentre retrospective study in low incidence countriesPLOS ONE Dear Dr. di Pietro,

Thank you for submitting your manuscript to PLOS ONE. After careful consideration, we feel that it has merit but does not fully meet PLOS ONE’s publication criteria as it currently stands. Therefore, we invite you to submit a revised version of the manuscript that addresses the points raised during the review process.

Major revision. The reviewers have recommended the possibility of a publication, but suggest some revisions to your manuscript. I kindly invite you to respond to the reviewers' comments and revise your manuscript accordingly.

We look forward to receiving your revised manuscript.

Kind regards,

Ozlem Boybeyi-Turer

Academic Editor

PLOS ONE

Journal Requirements:

Reviewers' comments:

Reviewer's Responses to Questions

**Comments to the Author**

1. Is the manuscript technically sound, and do the data support the conclusions?

Reviewer #1: Partly

Reviewer #2: Yes

2. Has the statistical analysis been performed appropriately and rigorously? 

Reviewer #1: No

Reviewer #2: Yes

3. Have the authors made all data underlying the findings in their manuscript fully available?

Reviewer #1: No

Reviewer #2: No

4. Is the manuscript presented in an intelligible fashion and written in standard English?

Reviewer #1: Yes

Reviewer #2: Yes

5. Review Comments to the Author

Reviewer #1: Manuscript title:

Adequacy of endoscopic recognition and surveillance of gastric intestinal

metaplasia and atrophic gastritis: a multicentre retrospective study in low

incidence countries (Manuscript ID: PONE-D-23-03805).

Authors:

J.Honing, W.Keith Tan, E. Dieninyte, M.O’Donovan, L.A.A.Brosens, B.L.A.M.Weusten,6 and M. di Pietro.

________

GENERAL COMMENTS

This study aims "to evaluate endoscopic recognition and adequacy of surveillance for GA/GIM in countries with low GAC prevalence."

Three centers from Netherlands and UK (years 2012- 2019) were retrospectively involved.

A consistent number of patients (396, with a median follow-up of 57.2 months) with GIM/GA diagnosis at index endoscopy were retrieved through a systematic search of pathology databases.

By assuming histology as the reference standard (?, if so, the histology criteria should be mentioned), endoscopic recognition rates were 48.5 % for GA and 16.3 % for GIM.

According to the ESGE guidelines, surveillance was adequately carried out in 215 out of 396 patients (54.3%).

The Authors (wisely) conclude that in countries with a low GC incidence, the rate of endoscopic recognition of gastric pre-cancerous lesions and adherence to surveillance recommendations are low.

The issue addressed by this study is of primary importance, that's why my personal opinion about the priority of this study. The study, however, includes some major methodological weaknesses that should be carefully addressed to make the Authors' conclusions consistent with an evidence-based approach.

In retrieving the considered patients, the authors regarded the definitions of atrophic gastritis and intestinal metaplasia as equivalent. The two conditions are not biologically nor clinically the same, being IM one of the histological phenotypes included in the Atrophy definition.

The reliability of endoscopy in atrophy detection is considered lower than that of IM (particularly by adopting high-resolution instruments). It is frankly unexpected the higher value of consistency in endoscopic detection of atrophy versus IM (48.5 % for GA versus 16.3 % for IM). This impressive result should be carefully considered and confirmed.

The definition of pangastritis includes heterogeneous clinical and histological situations. According to the Sydney system, the current (ambiguous) definitions of pangastritis includes different conditions associated with different GC risk:

(a) sparse foci of atrophy involving the mucous-secreting antrum and the oxyntic compartment;

(b) extensive atrophic lesions involving both the antrum and body (i.e., Open Type gastritis according to Kimura and Takemoto).

I agree entirely with the author's choice of assuming histology as a reference standard in atrophy assessment. Histology reliability - however - is strongly conditioned by the biopsy sampling protocol. No information about this crucial issue is reported.

All those mentioned above "basic" discrepancies in gastritis classification/definition may result in an equivocal clinical setting which may ultimately affect both the choice of the follow-up strategy and the consistency of the follow-up schedule with the considered ESGE guidelines.

Reviewer #2: The paper titled: "Adequacy of endoscopic recognition and surveillance of gastric intestinal metaplasia and atrophic gastritis: a multicentre retrospective study in low incidence countries" is very interesting regarding the rate of endoscopic recognition of gastric pre-cancerous lesions. Also, the authors state well the limitations of this study. However, I would like to ask the authors to provide more info with regards to the histological confirmation of the cases that they studies. For all the cases did the histological analysis confirm correctly the status of each case? Also, what was the time between the histological analysis and the endoscopy. The authors should provide a comparison between endoscopic and histological confirmation of the cases (as the histological confirmation remains the gold standard). Finally, a further discussion is needed with regards to the AI tools for the confirmation of these cases. For example, I would suggest the authors take into account and discuss the following recent work: "A digital pathology workflow for the segmentation and classification of gastric glands: Study of gastric atrophy and intestinal metaplasia cases".

Generally, I think that this article will certainly be of interest to many readers of PLOS One.

6. PLOS authors have the option to publish the peer review history of their article (what does this mean?). If published, this will include your full peer review and any attached files.

Reviewer #1: No

Reviewer #2: No

---

## [Author Response · Author response to Decision Letter 0]

16 May 2023

We have added the dataset and addressed the styling requirements.

---

## [Decision Letter · Decision Letter 1]

8 Jun 2023

Adequacy of endoscopic recognition and surveillance of gastric intestinal metaplasia and atrophic gastritis: a multicentre retrospective study in low incidence countries

PONE-D-23-03805R1

Dear Dr. di Pietro,

We’re pleased to inform you that your manuscript has been judged scientifically suitable for publication and will be formally accepted for publication once it meets all outstanding technical requirements.

Kind regards,

Ozlem Boybeyi-Turer

Academic Editor

PLOS ONE

Additional Editor Comments (optional):

Congratulations for your manuscript since the authors have addressed and clarified all required points. Therefore, I think the revised version of the manuscript is suitable for publication in PLOSONE.

Reviewers' comments:

Reviewer's Responses to Questions

**Comments to the Author**

1. If the authors have adequately addressed your comments raised in a previous round of review and you feel that this manuscript is now acceptable for publication, you may indicate that here to bypass the “Comments to the Author” section, enter your conflict of interest statement in the “Confidential to Editor” section, and submit your "Accept" recommendation.

Reviewer #2: All comments have been addressed

2. Is the manuscript technically sound, and do the data support the conclusions?

Reviewer #2: Yes

3. Has the statistical analysis been performed appropriately and rigorously? 

Reviewer #2: Yes

4. Have the authors made all data underlying the findings in their manuscript fully available?

Reviewer #2: Yes

5. Is the manuscript presented in an intelligible fashion and written in standard English?

Reviewer #2: Yes

6. Review Comments to the Author

Reviewer #2: In the revised manuscript, the authors have provided a more thorough explanation of their methodology, addressing the gaps and clarifying any ambiguities that were present in the initial version. The additional details and clarity have improved the understanding of their research approach.

7. PLOS authors have the option to publish the peer review history of their article (what does this mean?). If published, this will include your full peer review and any attached files.

Reviewer #2: No

---

## [Editor Report · Acceptance letter]

14 Jun 2023

PONE-D-23-03805R1 

Adequacy of endoscopic recognition and surveillance of gastric intestinal metaplasia and atrophic gastritis: a multicentre retrospective study in low incidence countries 

Dear Dr. di Pietro:

I'm pleased to inform you that your manuscript has been deemed suitable for publication in PLOS ONE. Congratulations! Your manuscript is now with our production department. 

Kind regards, 

on behalf of

Professor Ozlem Boybeyi-Turer 

Academic Editor

PLOS ONE